# From inpatient to outpatient mental health care: Protocol for a randomised feasibility trial of a care transition intervention for patients with depression and anxiety (the AMBITION-trial)

**Justus Tönnies, Marayah Ayoub-Schreifeldt, Viola Schrader, Mechthild Hartmann, Beate Wild, Hans-Christoph Friederich, Markus W. Haun** * 

Department of General Internal Medicine and Psychosomatics, Heidelberg University, Heidelberg, Baden-Württemberg, Germany

* markus.haun@med.uni-heidelberg.de

## Abstract

### Introduction

Despite guideline recommendations, inpatients with mental health disorders often do not receive appropriate treatment after discharge. This leads to high readmission rates, problems with medication adherence, increased risk of chronicity and suicide, and exclusion from the labour market accompanied by high individual and social costs. The causes are both system-related, such as limited treatment availability, and patient-related, such as ambivalent motivation to continue treatment and lack of information about available treatment options. The aim of this trial is to assess the feasibility of a Care Transition Intervention (CTI) which supports patients in the psychosocial follow-up treatment process after discharge from a psychotherapy ward.

### Methods and analysis

Fifty patients with depression and/or anxiety who are treated as inpatients at a psychotherapy ward will be included and randomised into two groups with a 1:1 ratio. In the intervention group, patients will receive five CTI sessions with a Care Transition Navigator before and after discharge. The sessions will focus on individual patient support including a) identification and tackling of barriers to initiate follow-up treatment, b) reflection on the inpatient stay and individual progress, with focus on the helpful aspects and c) motivation of patients to organise and take up outpatient treatment. Patients in the control group will receive treatment-as-usual during discharge. We will evaluate the following outcomes: effectiveness of recruitment strategies, patient acceptance of randomisation, practicability of implemented workflows, feasibility of data collection, and clinical outcomes.

**Data Availability Statement:** No datasets were generated or analysed during the current study. All relevant data from this study will be made available upon study completion.

**Funding:** This study (MW, MAS, and VS) supported by the Central Research Institute of Ambulatory Health Care in Germany (Grant number: D.10054158). The funders did not and will not have a role in study design, data collection and analysis, decision to publish, or preparation of the manuscript.

**Competing interests:** The authors have declared that no competing interests exist.

# Introduction

For patients with mental illness, the transition from inpatient to outpatient care is crucial. However, it is at this point that the continuity of care is often interrupted [1–3]. Patients have issues finding their way in the commonly indicated outpatient follow-up treatment. This leads to several negative effects such as; high readmission rates, problems with medication adherence, increased risk of chronicity and suicide, and exclusion from the labour market accompanied by high individual and social costs [1, 3–8]. Reasons for not starting outpatient follow-up treatment are on a structural (low treatment capacities of mental health specialists) as well as individual level (ambivalent motivation towards further treatment, lack of access to information about available treatment options and resulting difficulties in navigating through the health care system [9–11]).

While the barriers at the individual level appear obstructive, they are modifiable in the short term [12]. Patients often need support to prepare for the time after discharge and to orient themselves in the complex system of outpatient psychosocial care. This is where Care Transition Interventions (CTIs) come in, by providing close support for individual patients at an early stage, i.e. before and shortly after discharge. CTIs can include coordination between the different service providers (integrated care) and motivate patients to seek follow-up treatment independently by taking a closer look at their disorder (patient emancipation through psychoeducation) [13]. Systematic reviews on CTIs show a potential benefit for patients regarding risk of readmissions and symptom severity [14–17].

However, the literature shows that due to the high variability of a) intervention components, b) outcome measures, c) target groups/conditions, and d) methodological quality of the studies, there is not enough evidence to make a final recommendation for the use of CTIs. In addition, most of the included studies were conducted in the USA, where funding, organisational and health care structures differ from those in many other countries, e.g. those in Central Europe [18]. It is also not yet evident whether CTIs are cost-effective. The few studies that included cost-effectiveness analyses came to inconclusive and/or non-significant results. This is partly because the individual interventions were designed very differently (e.g. internet-based programme, peer support, clinician support) [19–22]. Nevertheless, within these broad variations of interventions components seem to be consistently effective. Therefore, we combined these components and developed a CTI to support patients with depressive and/or anxiety disorders at the transition from inpatient to outpatient psychosocial care.

To prepare the ground for a future effectiveness trial, we will conduct a randomised trial assessing the feasibility of the study procedures and the acceptability of a CTI conducted by a Care Transition Navigator (CTN) within the German health care system.

## Methods and analysis

### Study setting

The main setting of this feasibility trial are German hospitals in Southwest Germany (Federal State of Baden-Wuerttemberg). Inpatients will be recruited from mental healthcare wards and day clinics in German hospitals, where, patients with a wide range of mental disorders, from depression and anxiety disorders to eating disorders and obsessive-compulsive disorders receive treatment with a focus on intensive psychotherapy.

### Study design

This study is an individually randomised feasibility trial. After the inclusion of patients, the individual intervention period will be three months; for the total recruitment time 6 months

are planned. There will be two main measurements; a baseline assessment just prior to randomisation and a post assessment three months after inclusion. There will be two follow-up measurements at six and nine months after inclusion which focus on the medium-term effects. Patient recruitment August 2022 and will probably be completed in September 2023. Data collection will probably be completed by May 2024. The study protocol is reported and implemented according to Standard Protocol Items: Recommendations for Interventional Trials (SPIRIT) guidelines (S1 File) [23].

## Inclusion and exclusion criteria

CTIs have not been evaluated or even implemented in German routine mental healthcare. Indeed, in reviews on transitional interventions not a single trial from Germany was included [15, 16]. Hence, we selected depression and anxiety as highly prevalent mental health disorders with high burden of disease to pilot the CTI in German mental healthcare. Inclusion criteria require inpatients to 1) have a recommendation by the case leading therapist on the ward to start an outpatient psychosocial follow-up treatment, 2) be treated for at least a moderate depressive and/or anxiety disorder (additional comorbid mental health disorders allowed), 3) agree to participate in the study by written informed consent, 4) be capable of giving consent and 5) be 18 years or older. Exclusion criteria are as follows: 1) planned resumption of outpatient psychotherapy after the inpatient stay which has been initiated before admission, 2) acute psychotic symptoms, e.g., persecutory delusions and/or thought insertion, 3) severe cognitive impairment or dementia, 4) significant hearing and/or visual impairment, 5) pregnancy in the $\geq$ 2nd trimester, and 6) insufficient German language proficiency. For this pilot trial, we aimed for minimizing the rate of loss-to-follow-up. Thus, we excluded pregnant women who may not have been able to complete follow-up assessments due to giving childbirth. We will drop this exclusion criterion in the main trial. Before enrolling patients, we will regularly check with providers having delivered inpatient treatment if a therapeutic dialogue was possible and whether language proficiency had turned out as a problem during inpatient treatment. To ensure the correct assessment of inclusion criteria, we will work closely with the case leading therapist on the respective ward / day clinic to record the recommendation for follow-up treatment and the diagnosis for which the patient was admitted. All other inclusion and exclusion criteria will be assessed by a study team member as part of the initial screening process.

## Study procedures

Prior to patient recruitment, we will meet with the hospital staff and introduce the study. We will precisely outline, how each member of the ward team will be involved and how they will guide the patients through the study during the inpatient stay. Additionally, during the preparatory visit we will address remaining questions and provide the team with a study handbook that includes a comprehensive description of patient inclusion criteria. The case leading therapists on the respective ward / day clinic will then start recruiting patients by introducing the study to them during regular therapy sessions. We anticipate that the therapeutic alliance with the case leading therapist will promote patients' willingness to participate in the study. If a patient is interested in participating, they will provide written informed consent and fill out the baseline questionnaire. Then patient's names will be pseudonymized and patients will be randomly allocated to the intervention or the control condition. The study flow is depicted in Fig 1 and the study schedule is shown in Fig 2.

Before the intervention starts, the CTN will attend a two hour-training session, during which they will be familiarised with strategies and practicalities pertaining to organising follow-up treatments for patients with depression and anxiety. We will also train the CTN in

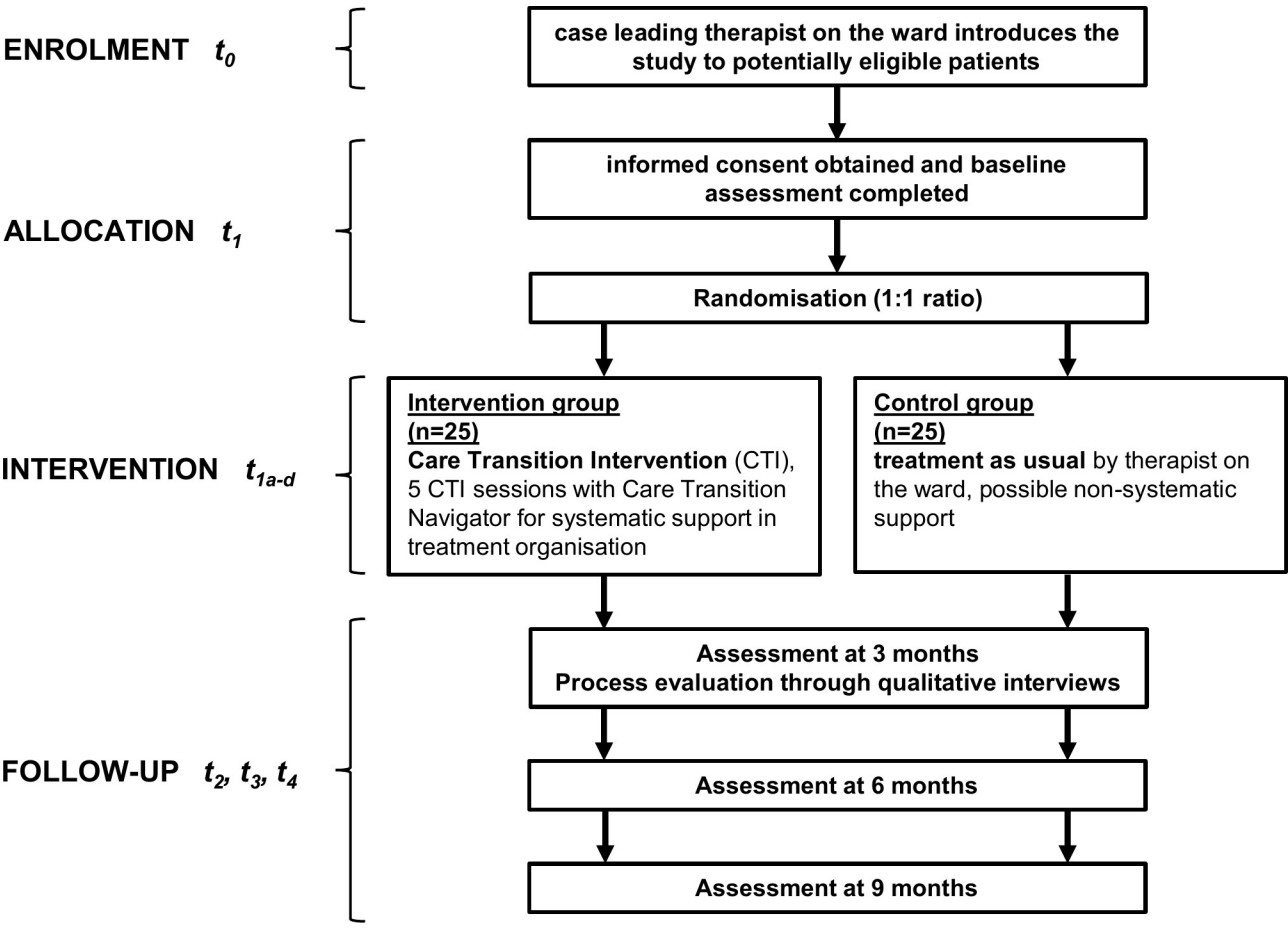

**Fig 1. Study flowchart.**

motivational interviewing and psychoeducation. Using motivational interviewing, we aim to encourage and empower the patients to take up follow-up treatment.

## Study conceptual framework

The AMBITION intervention targets the transition between an inpatient / day clinic stay and an outpatient follow-up treatment (Fig 3) and accounts for the five domains for factors influencing treatment adherence established by the World Health Organization (WHO): social and economic, healthcare system, therapy-related, patient-related, and condition-related [28]. The intervention unfolds across these factors by tightening coordination between inpatient and community mental healthcare.

We developed the CTI based on a thorough review of the literature (MEDLINE and Google Scholar with the search term "mental health treatment engagement" and no filters) finalized on February 8th, 2022. The literature was reviewed by a group of experts of different backgrounds (psychiatrist, epidemiologist, psychologist). Specifically, we selected the intervention components with respect to their potential effectiveness for increasing mental health treatment engagement accounting for the five WHO domains for factors influencing treatment engagement and adherence (see Table 1 for details).

| | Enrolment | Allocation | STUDY PERIOD | | | | | | | Close-out |
|---|---|---|---|---|---|---|---|---|---|---|
| | | | Post-allocation | | | | | Follow-up | | |
| **TIMEPOINT** | $t_0$ | $t_1$ | $t_{1a}$ | $t_{1b}$ | $t_{1c}$ | $t_{1d}$ | $t_2$ ($t_1+3$ months) | $t_3$ ($t_1+6$ months) | $t_4$ ($t_1+9$ months) | $t_x$ |
| **ENROLMENT:** | | | | | | | | | | |
| *case leading therapist on the ward approaches suitable patient* | X | | | | | | | | | |
| *informed consent* | | X | | | | | | | | |
| *baseline assessment* | | X | | | | | | | | |
| *randomisation* | | X | | | | | | | | |
| *allocation* | | X | | | | | | | | |
| **INTERVENTIONS:** | | | | | | | | | | |
| *Care Transition Intervention (CTI)* | | | ◆━━━━◆ | | | | | | | |
| *treatment as usual* | | | ◆━━━━◆ | | | | | | | |
| **ASSESSMENTS:** | | | | | | | | | | |
| *sociodemographic data* | | X | | | | | | | | |
| *PHQ-ADS[1], SF-12[2], RAS-G[3], SSD-12[4]* | | | | | | | X | X | X | |
| *interviews: CTN, patients from the interventions group* | | | | | | | X | | | |

**Fig 2. Study schedule.** [1] Patient Health Questionnaire Anxiety and Depression Scale (PHQ-ADS) [24], [2] 12-Item Short-Form Health Survey (SF-12) [25], [3] Recovery Assessment Scale (RAS-G) [26], [4] Somatic Symptom Disorder–B Criteria Scale (SSD-12) [27].

## Intervention

We will not provide psychotherapy focused on direct symptom improvement during the intervention aim of the CTI is to empower and support the patient to take up follow-up treatment (e.g. seeking a guideline-based adequate follow-up care in the form of remission-stabilising maintenance therapy). The CTI will be delivered by a CTN and is composed of core components and optional components. Core components have been shown to be effective in increasing mental health treatment engagement in previous work, which will be the primary outcome in sufficiently powered confirmatory trial after this pilot. Hence, core components are compulsory parts of the intervention. The optional components will be used as needed by the CTN to

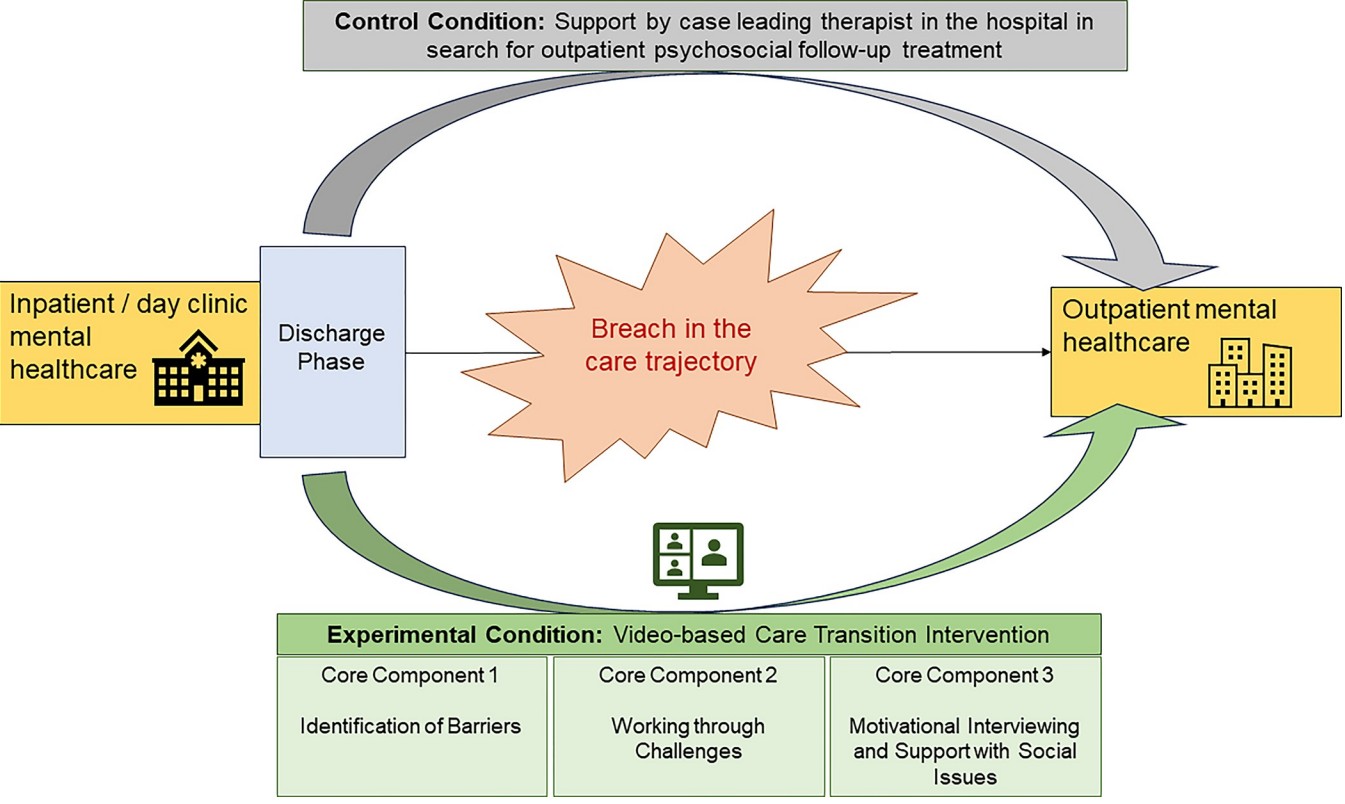

**Fig 3. Schematic illustrating the care transition intervention and treatment-as-usual.**

provide more individualised guidance and support to each patient. The type and design of the outpatient psychosocial follow-up treatment will primarily be based on the recommendations of the case leading therapist on the ward / day clinic. The first core component is the identification and subsequent addressing of barriers that prevent the patient from taking up an outpatient psychosocial follow-up treatment [29, 30]. The CTN will use motivational interviewing and psychoeducation to encourage the patient to make and keep appointments with outpatient mental health services. The patient, supported by the CTN, will develop a summary of the inpatient stay from her or his own perspective. This enables the CTN to identify both challenges and achievements from the patient's point of view and to address the respective challenges more precisely in further joint work that includes emotional support for the patient (core component 2) [31]. For example, the type of outpatient follow-up treatment can be adapted based on factors which have already contributed to improvements for the patient during their inpatient stay. In preparation for the intervention, the CTN will receive extensive study-related training in motivational interviewing, diversity, and gender-specific healthcare-seeking behaviours and social work topics (core component 3) [32, 33]. The CTN will be a research assistant with a bachelor's degree in psychology. These steps will take place during the first joint appointment (CTI session) before the patient is discharged.

After discharge, further CTI sessions will take place between the patient and the CTN, during which barriers on the patient's side will be continuously identified and addressed. Additionally, if necessary the CTN will reflect on the ambivalence towards follow-up treatment with the patient, to build motivation. In the first month after discharge, there will be two CTI sessions; in the months two and three after discharge, there will be one CTI session per month.

**Table 1. Study conceptual framework.**

| WHO Domain for Factors Influencing Treatment Engagement and Adherence | AMBITION Intervention | |
|---|---|---|
| | **Specific Targets** | **Strategies or techniques** |
| **Social and Economic** | Socioeconomic status | Readily accessible technology-facilitated intervention |
| | Employment and education | Low-threshold intervention compatible with work- and family-related obligations |
| | Health literacy | Use of well comprehensible educational material and graphical instructions |
| | Social support | Provision of emotional support |
| | Treatment costs | Intervention at no cost (covered by funding in the pilot stage) |
| | Racial, ethnic, gender, and cultural background | Attentiveness for diversity and gender-specific healthcare-seeking behaviours |
| **Healthcare System** | Shortage of community mental health services | Active support in seeking treatment |
| | Fragmentation of care | Intervention at the time of great risk for discontinuity of care |
| | Patient-centred care | Fostering informed conversations between providers and patients |
| **Therapy-related** | Complexity | Psychoeducation on potential types of treatment and treatment duration |
| | Previous treatment failures | Exploration of previous treatment experiences (e.g., summary of inpatient / day clinic stay) |
| | Unintended consequences and adverse effects | Exploration of potential treatment adverse effects in the past |
| **Patient-related** | Health beliefs | Deepening the understanding for the respective mental health disorder |
| | Perceived barriers | Systematic identification and optimised removal of barriers |
| | Treatment engagement-related needs | Behavioural strategies for improving treatment engagement |
| **Condition-related (depressive disorders)** | Feelings of hopelessness | Fostering change expectancy |
| | Loss of interest and/or pleasure | Behavioural activation |
| | Rumination | Behavioural activation |

If a patient successfully takes up outpatient follow-up treatment before the fifth CTI session, at least one further CTI session will be offered for joint final reflection and assessment of the sustainability of follow-up treatment. If the patient needs more structuring of the intervention, a patient-centred written brief discharge plan will be jointly developed by the patient and the CTN. This will help to (1) clarify the need and (2) the desired design of a follow-up treatment, (3) identify possible barriers, (4) identify patient's areas/problems in particular need of support by the CTN, and (5) schedule the session dates between patient and CTN (1st month two sessions, 2nd and 3rd month one session each). To support and motivate the patient, the CTN can screen for psychosocial services and treatment possibilities in the vicinity of the patient's home. Furthermore, the patient will be motivated to contact the general practitioner after discharge and to inform him/her about the further course of treatment. Further components for support and motivation can vary individually depending on the symptom severity and the resources of the patient, but will include components of the following spectrum: systematic reminders about doctor's appointments and medication intake, screening for suicidal ideation, evaluation of already known and/or newly emerging barriers, psychoeducation regarding the handling of mental disorders and coping strategies and behavioural activation (maintaining a positive attitude, solution orientation, stimulation of self-help). These optional components have been shown to be effective in several comparable studies [14, 17]. The CTI sessions will be conducted via a secure (i.e., encrypted), web-based videoconferencing platform on a subscription basis (arztkonsultation ak GmbH) and will last between 30 and 60 minutes.

Subscription fees are covered by the trial funding. No fee will apply for patients. In routine care in Germany, the use of teleconferencing systems is covered by the statutory health insurance. During the intervention period, the CTN will attend a case supervision session led by a senior physician every two weeks, where all patients will be discussed. In particular, it will be discussed how to deal with hard-to-reach and less adherent patients to successfully implement the transition to outpatient psychosocial follow-up treatment in this group as well.

## Control condition

Patients allocated to the control group will receive the usual support by their case leading therapist on the ward in their search for outpatient psychosocial follow-up treatment (e.g. by being given contact details of outpatient psychotherapists in the vicinity and motivated to make contact during the further course of inpatient treatment). This will also include participating in a follow-up supportive group provided by the inpatient staff (five sessions for each patients) after patients have been discharged. This open group meets once a week to discuss problems in organising follow-up treatment and to provide rather general than individualised support.

## Sample size

In this study, we aim to establish feasibility of a full trial and not to provide evidence of a statistically significant difference concerning effectiveness between the two treatment conditions. Therefore, we did not perform a formal sample size calculation [34–36]. Instead, our target sample size of $N = 50$ participants is based on the recommendations for conducting pilot and feasibility trials as described by the National Institute for Health Research [37].

## Recruitment

Patients will be recruited during their inpatient / day clinic stay in mental healthcare departments of German Hospitals. Suitable patients will be approached by their case leading therapist on the ward / day clinic during the regular therapy sessions three weeks prior to discharge. Here they will be informed about the study and provided with the study documents, consisting of information leaflet and the consent form in duplicate. If the patient agrees to receive further information, they will have a meeting with the CTN two weeks prior to discharge. During this meeting the CTN will 1) describe the study procedures to the patient, 2) address any remaining questions the patient may have, 3) obtain written informed consent, 4) collect the patient's baseline data (either by filling out the questionnaire or through an online assessment), 5) allocate the patient to one of the two study conditions via an online randomisation platform, 6) inform the patient of the outcome of the randomisation, and, 7) if randomised to the intervention condition, arrange a first appointment (CTI session), which must still take place during the inpatient stay. If inclusion in the study is not possible, this will be documented with the reason and the patient's age and gender. The CTN informs the case leading therapist on the ward of the outcome result of the randomisation.

## Randomisation

After obtaining informed consent, patients will be randomly allocated to one of the two study conditions (treatment-as-usual, TAU vs. care transition intervention) in a 1:1 ratio. To ensure a seamless inclusion process, the CTN will conduct the randomisation during the initial recruitment meeting with the patient and subsequently inform the patient of the outcome. Concealment of the treatment sequence up to allocation will be assured via a secure web-based randomisation system (Randomizer, https://randomizer.at) operated by a data manager, not

involved in the patient recruitment, centrally at the Department of General Internal Medicine and Psychosomatics, Heidelberg University. The treatment sequence is generated through a computer-generated sequence of random numbers. Randomisation will be stratified by depression and anxiety symptom severity at baseline as measured with the PHQ-ADS (three levels: minimal/mild, moderate, severe).

## Measurements

This study will assess the feasibility and acceptance of both the CTI and study procedures, such as data collection, as it would occur in a sufficiently powered effectiveness trial. We will, therefore, evaluate if the procedures are appropriate for this specific patient group as well as collect data on clinical endpoints.

## Patients' health status

For patients, the baseline assessment will take place right before randomisation. This includes variables on sociodemographic information, patient's history of outpatient as well as inpatient mental health treatment and history of psychopharmacotherapy. Data will be collected using the validated questionnaires: Patient Health Questionnaire Anxiety and Depression Scale (PHQ-ADS) [24], 12-Item Short-Form Health Survey (SF-12) [25], Recovery Assessment Scale (RAS-G) [26], Somatic Symptom Disorder–B Criteria Scale (SSD-12) [27]. Since the aim of this study is to enable patients to organise their treatment following an inpatient stay and ideally take up outpatient psychosocial treatment, we will comprehensively record service use regarding such treatment options. Post and follow-up measurements will be conducted three, six and nine months respectively after study inclusion and will include the same validated questionnaires and service use assessments as at baseline. At this pilot stage, we will not account for the intensity of the treatment received prior to enrolment, since we are primarily interested in feasibility outcomes. However, in the sufficiently powered main trial, we will assess this variable applying the Questionnaire for the Assessment of Medical and non-Medical Resource Utilisation in Mental Disorders (FIMPsy) [38]. Research assistants will conduct the post and follow-up measurement in Computer Assisted Telephone Interviews (CATIs) with the participants. The study schedule is depicted in Fig 3 in line with the SPIRIT guidelines.

## Feasibility

In order to evaluate feasibility from the participants' perspective, we will conduct semi-guided qualitative interviews with patients from the intervention group as well as with the CTN. The acceptability of both study and intervention procedures will be evaluated by assessing whether the intervention and study procedures have been agreeable as well as logistically and technically practical [39]. A semi-guided interview with the CTN will give a more detailed insight into provider's perspective regarding feasibility and appropriateness of the CTI.

## Outcomes

As this is a feasibility trail, there will be no hypotheses or statistical testing. Therefore, effects of the intervention are not to be expected in this small sample. Instead, we expect that a) our trial design (incl. study procedures such as recruitment strategy, data collection procedures, randomisation and logistic aspects) is suitable and b) that the intervention is feasible in this population of inpatients and at this specific time point during treatment, the transition from inpatient to outpatient care.

The feasibility of a subsequent large-scale RCT for patients at the transition from inpatient to outpatient care will be determined by assessing the following outcomes and aspects [34]:

- recruitment strategy's sufficiency and efficiency for both groups

- adherence for intervention group

- feasibility of study procedures (e.g., patient and provider acceptance of randomisation and outcome measurements)

- intervention procedure feasibility (CTN documentation, patient acceptance of CTI, patient safety)

To estimate the sufficiency and efficiency of the recruitment strategies, we will calculate the recruitment yield, i.e., number randomized per number screened, and the consent rate, i.e., number randomized per number eligible. We will record retention rates and reasons for non-participation or dropping out. Adherence will be determined by the ratio of scheduled and actually conducted CTI session. Concerning documentation, after every CTI session, the CTN will systematically document which elements of the intervention were used in the respective session. Patients experience relapses will have full and free access to all levels of mental health-care which covered under the statutory health insurance in Germany. We will approach all patients systematically for post assessments regardless of whether they have received a relapse to minimize missing data for the follow-up. To evaluate process outcomes on the overall practicability of the intervention and the related study procedures, we will draw on qualitative data generated by in-depth interviewing of patients and the CTN.

## Data analysis

In order to assure data quality, we will 1) use a password protected online survey tool (Enterprise Feedback Suite (EFS) Survey, Questback GmbH) during the data collections and enter data from the paper questionnaire there, 2) enforce data integrity using forced or multiple-choice items wherever possible, and 3) ensure that only two members of the study team will have access to the data as well as the participants identifying information and will prepare it prior to data analysis. Quantitative data on clinical outcomes along with information on health service use will be analysed using descriptive statistics (absolute and relative frequencies, measures of central tendency, and measures of variability). We will not test any hypotheses or perform any statistical tests, but rather will assess the feasibility of a following large-scale RCT. We will report various dimensions regarding feasibility, e.g., overall recruitment yield (number randomised per number screened), the recruitment rate (number recruited and randomised per general practice per month), consent rate (number randomised per number eligible), and loss to follow-up. To illustrate participant flow, we will report results in a CONSORT diagram [40]. We will describe patients' reasons for non-participation as well as conduct a non-responder analysis. Qualitative data stemming from the process evaluation (i.e., interviews with patients and CTN) will be analysed utilizing the thematic analysis approach deriving key themes bottom-up. For the descriptive quantitative analysis, we will use R (version 4.3.1 or higher).

To gain a vaster understanding of the context in which the outcomes of the RCT developed, we will conduct a process evaluation consistent of qualitative semi-guided interviews with the patients and CTNs [41]. We will purposefully sample participants concerning hard-to-reach groups (oversampling men and patients with chronic and multimorbid conditions) The interviews will focus on barriers as well as facilitators of the model. Process data will be collected after the intervention, to gain insight into the various intervention stages [42]. The top-down

themes of for the thematic analysis of the interview data will be 1) quality of implementation, (2) causal mechanisms/pathways, and (3) contextual influences on the implementation of the intervention. Coders executing the qualitative analysis will not be part of the delivery of the intervention nor analyze the qualitative data prior to knowing the trial outcomes.

For the thematic analysis of the qualitative data we will use MAXQDA 2020 or higher. All study publications will align with recommendations of the respective statements for randomised trials (CONSORT [40]) and qualitative research [43].

## Missing values

During the data collection process, we will try to keep the number of missing values at a minimum by carrying out most assessments during same-room meetings or by making use of CATIs (i.e. post measurements and follow-up). To reduce missing values during the collection of baseline-data, we will assist patients where necessary, and check the completeness of each patients' baseline questionnaire before randomisation. We will consider the occurrence of missing values despite these two procedures as important findings in the context of this feasibility trial. This could bring forth important information on the acceptance and appropriateness of the questionnaires used.

## Patient and public involvement

This protocol was drafted without patient involvement and patients were not invited to contribute to the writing or editing of this document for readability or accuracy. However, as part of the feasibility and acceptance assessment we will conduct semi-structured interviews with all patients in the intervention condition. Interviews will focus on five main questions (What was your experience with the care transition intervention like? What worked out for you? What did not go well? Would you engage in the intervention again and/or recommend it to others? Why? Why not? Do you have any suggestions for improving the intervention? Which logistical problems did you experience? If applicable, how were they dealt with?). At the end of the interview, we will invite all patients to be systematically included in the refinement of the intervention. This could include development of the procedures and materials of a consecutive sufficiently powered effectiveness trial. We will align patient and public involvement with current guidelines [44].

## Ethics and dissemination

Publishing pilot protocols ensures consistency between the published protocol and published objectives, methods, and results. Moreover, published pilot protocols increase the transparency concerning details of the planned study and the likelihood of collaboration between groups of researchers [45]. Only patients who have given their written consent by signing the corresponding form will be included in the study. They will receive detailed information about the study and their right to withdraw without giving reasons through an information sheet and a comprehensive introduction by the CTN, in advance. Afterwards they will have the possibility to ask the principal investigator any questions they may have. Before the intervention starts, the CTN will attend a training session, led by an experienced mental health specialist to be adequately prepared to conduct the intervention. Moreover, the CTN will be supported and guided by a biweekly supervision which will be led by a senior consultant both in general and psychosomatic medicine from the Department of General Internal Medicine and Psychosomatics, Heidelberg University. We do not expect major relevant risks for participants, since the intervention's aim is to support patients in the organisation of treatment following an inpatient stay rather than providing a psychotherapeutic intervention. Completing assessments

will be of reasonable time burden for the participants and they will have the possibility to receive assistance through the process. The data collection and storage will be conducted in accordance with the General Data Protection Regulation (GDPR) ensuring a high level of data safety and a conscientious handling of all patient data. Ethical approval for the study has been granted from the Medical Faculty of the University of Heidelberg Ethics Committee, Reference S-259/2022. We will publish and present key findings on conferences and in internationally recognised peer reviewed journals. This feasibility trial will prepare the ground for a sufficiently powered randomised trial. The trial registration number is: DRKS00029381.

## Discussion

Since the transition between the inpatient and outpatient mental health care sector is often a breaking point in the continuity of care, it is crucial to develop and evaluate interventions that help patients in this process. While we agree that some evidence for the effectiveness of TIs is available, particularly for veterans with serious mental illness treated within the VA system, our trial follows a new approach in that we apply a video-based approach that features (1) systematic identification of barriers for follow-up treatment, (2) highlighting patients' problem-solving resources, and (3) motivational interviewing to keep patients engaged in care. In this respect, our intervention is more structured und builds on evidence-based components compared to those implemented in previous work. In addition, our intervention uses relatively low resources by employing a research assistant as CTN rather than a clinician or social worker. This may provide a low threshold of implementation for institutions. Particularly, these "bridging" interventions, which include pre-discharge as well as post-discharge components have showed promising results [19, 46].

## Supporting information

**S1 File. SPIRIT-checklist.**
(DOCX)

**S2 File. PLOS ONE clinical studies checklist.**
(DOCX)

**S3 File. Ethics committee study protocol original.**
(PDF)

**S4 File. Ethics committee study protocol translation.**
(PDF)

**S5 File. Ethics committee approval original.**
(PDF)

**S6 File. Ethics committee approval translation.**
(PDF)

## Author Contributions

**Conceptualization:** Viola Schrader, Beate Wild, Hans-Christoph Friederich.

**Funding acquisition:** Markus W. Haun.

**Investigation:** Justus Tönnies, Viola Schrader, Beate Wild.

**Methodology:** Justus Tönnies, Beate Wild.

**Project administration:** Markus W. Haun.

**Resources:** Hans-Christoph Friederich.

**Supervision:** Mechthild Hartmann.

**Writing – original draft:** Justus Tönnies, Markus W. Haun.

**Writing – review & editing:** Marayah Ayoub-Schreifeldt, Viola Schrader, Mechthild Hartmann, Beate Wild, Hans-Christoph Friederich.

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
