## [Decision Letter · Decision Letter 0]

5 Jul 2023

PONE-D-23-11362From inpatient to outpatient mental health care: protocol for a randomised feasibility trial of a care transition intervention for patients with depression and anxiety (the AMBITION-trial)PLOS ONE

Dear Dr. Haun,

Thank you for submitting your manuscript to PLOS ONE. After careful consideration, we feel that it has merit but does not fully meet PLOS ONE’s publication criteria as it currently stands. Therefore, we invite you to submit a revised version of the manuscript that addresses the points raised during the review process.

We look forward to receiving your revised manuscript.

Kind regards,

Dickens Akena, Ph.D

Academic Editor

PLOS ONE

Journal Requirements:

- Haun MW, Tönnies J, Graue L, et alMental health specialist video consultations for patients with somatic symptom disorder in primary care: protocol for a randomised feasibility trial (the VISION trial)BMJ Open 2022;12:e058150. doi: 10.1136/bmjopen-2021-058150

In your revision ensure you cite all your sources (including your own works), and quote or rephrase any duplicated text outside the methods section. Further consideration is dependent on these concerns being addressed.

5. We note that the original protocol that you have uploaded as a Supporting Information file contains an institutional logo. As this logo is likely copyrighted, we ask that you please remove it from this file and upload an updated version upon resubmission.

**Additional Editor Comments:**

Good day. I have received comments from the reviewers.  The article needs some tweaking before it is ready for publication. The authors need to respond to each of the comments raised by all the reviewers

Kind regards

Dickens Akena,

Academic Editor

Reviewers' comments:

Reviewer's Responses to Questions

**Comments to the Author**

1. Does the manuscript provide a valid rationale for the proposed study, with clearly identified and justified research questions?

Reviewer #1: Yes

Reviewer #2: Yes

Reviewer #3: Partly

Reviewer #4: Yes

2. Is the protocol technically sound and planned in a manner that will lead to a meaningful outcome and allow testing the stated hypotheses?

Reviewer #1: Yes

Reviewer #2: Yes

Reviewer #3: Partly

Reviewer #4: Yes

3. Is the methodology feasible and described in sufficient detail to allow the work to be replicable?

Reviewer #1: Yes

Reviewer #2: Yes

Reviewer #3: No

Reviewer #4: Yes

4. Have the authors described where all data underlying the findings will be made available when the study is complete?

Reviewer #1: Yes

Reviewer #2: Yes

Reviewer #3: Yes

Reviewer #4: Yes

5. Is the manuscript presented in an intelligible fashion and written in standard English?

Reviewer #1: Yes

Reviewer #2: Yes

Reviewer #3: Yes

Reviewer #4: Yes

6. Review Comments to the Author

You may also provide optional suggestions and comments to authors that they might find helpful in planning their study.

Reviewer #1: Important note: This review pertains only to ‘statistical aspects’ of the study and so ‘clinical aspects’ [like medical importance, relevance of the study, ‘clinical significance and implication(s)’ of the whole study, etc.] are to be evaluated [should be assessed] separately/independently. Further please note that any ‘statistical review’ is generally done under the assumption that (such) study specific methodological [as well as execution] issues are perfectly taken care of by the investigator(s). This review is not an exception to that and so does not cover clinical aspects {however, seldom comments are made only if those issues are intimately / scientifically related & intermingle with ‘statistical aspects’ of the study}. Agreed that ‘statistical methods’ are used as just tools here, however, they are vital part of methodology [and so should be given due importance]. I look at the manuscript in/with statistical view point, other reviewer(s) look(s) at it with different angle so that in totality the review is very comprehensive. However, there should be efforts from authors side to improve (may be by taking clues from reviewer’s comments). Therefore, please do not limit the revision only (with respect) to comments made here.

COMMENTS: Although this manuscript is well drafted [and the study is excellent with respect to most of the aspects (This study being ‘pilot’ in nature, sample size is not a big issue anyway)], I have few observations/concerns (different opinion) which are given below:

Since (as you have mentioned) in ‘Introduction’ section that “Systematic reviews on CTIs show a potential benefit for patients regarding risk of readmissions and symptom severity [14–17]”, and later in ‘Intervention’ section you also state that “optional components have been shown to be effective in several comparable studies [14,17].”, I wonder if there is really a need of such a ‘feasibility trial’? [particularly because the quoted studies are published in reputed journals and after looking at the competence/knowledge of the topic/preparation of the authors]. They could have planned a larger main/final phase-III trial (as said sufficiently powered randomised controlled trial) as well, I guess.

In my knowledge the primary purpose of publishing a research protocol (of main study/trial) is a means to allow the academic community to evaluate whether subsequent analysis and results are in line with the investigators' initial objectives. Additionally, it informs the academic community on ongoing research and may avoid duplication of work. Agreed that making study protocols publicly available has the benefit of disseminating the most contemporary ideas with respect to study design and data analysis. However, the purpose/intention of ‘protocol for a pilot/feasibility study/trial’ is/are not known {at least to me and may be to many others as well}. Will these learned authors may please explain?

[Note that the statement “Patient recruitment will start in July 2022 and will probably be completed in July 2023” in this manuscript prompted me to write the above as this excellent study will get delayed].

Except these minor points, the article is acceptable. However, mind you that as pointed out in ‘important note’ above “This review pertains only to ‘statistical aspects’ of the study and so ‘clinical aspects’ should be assessed separately/independently. ‘Minor Revision’ is recommended.

Reviewer #2: The authors provide a protocol for their CTI trial aimed at enhancing use of mental health services after discharge. There are a few comments which could be considered;

1. It is not clear how many psychotherapy sessions the patients will have had prior to enrollment during admission. It is possible that varying numbers of sessions may affect the outcomes.

2. The core components of the intervention are mentioned. However there no studies provided to show that these components are essential for the outcomes they are measuring (for the optional components, some citations were provided).

3. The CTI sessions will be provided via a web based teleconference system that is paid for. Won't this be a barrier for some patients who may not have the system or how to use it?

4. Sufficiency and efficiency are some of the outcomes being measured. How will these two be analyzed?

Reviewer #3: The authors address an important aspect of mental health services and provide a rationale for bridging inpatient and outpatient care, however , the rationale for the chosen illness conditions (Depression and Anxiety) is not provided. This makes it difficult to appreciate why CTI (the intervention ) should be the intervention of choice.

Of note, CTI is a multicomponent intervention but the detail on how it has been developed , and what the effective components are (core and optional ) would help one to appreciate how CTI is likely to be effective. How will the patients be involved in the development of the intervention like is stated in the protocol (patient and public involvement) ?

A schematic or diagram illustrating CTI (and the treatment as usual for the control arm) would help explain the two study treatments under comparison.

A study conceptual framework should also be included in the protocol.

It is not clear to me how patient recruitment will start in July 2022 and end in 2023 yet the protocol is under review in July 2023; why is pregnancy in the 2nd and 3rd trimester part of the exclusion criteria? And how will insufficient German language proficiency be determined ?

With respect to the intervention; how will relapses be handled in the study?

How will the assessors who will collect data be blinded to the treatment allocation? this was not clear,

The study seems like it will use a mixed methods approach but there is limited detail on qualitative methods, specifically the interviews; do they mean in depth interviews with patients and key informant interviews with the CTN? More elaboration on this aspect of the study is required including the qualitative data collection and analysis .

Reviewer #4: The authors clearly and intelligibly wrote this manuscript in standard English. The rationale was stated well and it indeed is a worthwhile project. The protocol is technically sound and the methods used are replicable.

ISSUES to address

The links in the appendix on pages 30,31,32 and 33 all link to download of the SPIRIT guidelines. These should be redirected to the appropriate documents.

7. PLOS authors have the option to publish the peer review history of their article (what does this mean?). If published, this will include your full peer review and any attached files.

Reviewer #1: No

Reviewer #2: No

Reviewer #3: No

Reviewer #4: No

---

## [Author Response · Author response to Decision Letter 0]

15 Jul 2023

Review Comments to the Author

Reviewer #1:

Since (as you have mentioned) in ‘Introduction’ section that “Systematic reviews on CTIs show a potential benefit for patients regarding risk of readmissions and symptom severity [14–17]”, and later in ‘Intervention’ section you also state that “optional components have been shown to be effective in several comparable studies [14,17].”, I wonder if there is really a need of such a ‘feasibility trial’? [particularly because the quoted studies are published in reputed journals and after looking at the competence/knowledge of the topic/preparation of the authors]. They could have planned a larger main/final phase-III trial (as said sufficiently powered randomised controlled trial) as well, I guess.

AUTHORS’ REPLY: Thank you very much for pointing out this aspect. It is correct that there is some evidence that transitional interventions are effective with respect to risk reduction and other outcomes. However, only few studies evaluated the employment of a transition manager. Specifically, in the most thorough review on transitional interventions (TI) provided by Vigod et al. (2018; https://doi.org/10.1192/bjp.bp.112.115030) only five of the 15 studies included in the review implemented a transition manager: Two trials focussed on critical time interventions for veterans with serious mental illness (Dixon et al., 2009; Kasprow & Rosenheck, 2007), one very small trial (N = 13) on advanced practice psychiatric nurse communication with clients suffering from schizophrenia after discharge via prepaid cellular phones (Price 2007). Another very small trial (N = 25) investigated peer support and overlap of inpatient and community staff in “patients diagnosed with a range of mental illnesses” (Reynolds et al., 2004). Finally, Forchuk et al. (2005) evaluated overlap of in-patient and community staff and peer support in a sample with heterogeneous diagnoses.

While we agree that some evidence for the effectiveness of TIs is available, particularly for veterans with serious mental illness treated within the VA system, our trial by and large follows a new approach in that we apply a video-based approach that features (1) systematic identification of barriers for follow-up treatment, (2) highlighting patients’ problem-solving resources, and (3) motivational interviewing to keep patients engaged in care. In this respect, our intervention is more structured und builds on evidence-based components compared to those implemented in previous work. In the Discussion section on pages 21-22, we now state what our study will likely add to the existing literature.

However, the purpose/intention of ‘protocol for a pilot/feasibility study/trial’ is/are not known {at least to me and may be to many others as well}. Will these learned authors may please explain?

[Note that the statement “Patient recruitment will start in July 2022 and will probably be completed in July 2023” in this manuscript prompted me to write the above as this excellent study will get delayed].

AUTHORS’ REPLY: Publishing protocols of pilot and feasibility trials has been advocated as an important strategy towards improving transparency (Thabane, 2019; https://doi.org/10.1186/s40814-019-0423-8). Specifically, it is argued that publishing pilot protocols avoids selective reporting the results, ensures consistency between the published protocol and published objectives, methods, and results, increases transparency concerning details of the planned study, improves the methodological quality of the main study, and raises investigators’ accountability. Published protocols of pilot studies also make it more likely that groups of researchers collaborate and expand the planned main study. We briefly point out these advantages of publishing protocols of pilot trials in the Discussion section on page 20.

Except these minor points, the article is acceptable. However, mind you that as pointed out in ‘important note’ above “This review pertains only to ‘statistical aspects’ of the study and so ‘clinical aspects’ should be assessed separately/independently. ‘Minor Revision’ is recommended.

AUTHORS’ REPLY: We are pleased that Reviewer #1 finds our work appealing and thank them for their time.

Reviewer #2:

There are a few comments which could be considered;

1. It is not clear how many psychotherapy sessions the patients will have had prior to enrollment during admission. It is possible that varying numbers of sessions may affect the outcomes.

AUTHORS’ REPLY: Thank you for pointing this out. At this pilot stage, we will not account for the intensity of the treatment received prior to enrolment, since we are primarily interested in feasibility outcomes. However, in the sufficiently powered main trial, we will assess this variable applying the FIMPsy – Questionnaire for the Assessment of Medical and non-Medical Resource Utilisation in Mental Disorders (Grupp et al., 2018; https://doi.org/10.1055/s-0042-118033). We have added this to the Measurements subsection on page 16.

2. The core components of the intervention are mentioned. However there no studies provided to show that these components are essential for the outcomes they are measuring (for the optional components, some citations were provided).

AUTHORS’ REPLY: Core components of the intervention are: (1) identification and subsequent addressing of barriers that prevent the patient from taking up an outpatient psychosocial follow-up treatment, (2) working through challenges and achievements from the patient’s point of view, (3) training in motivational interviewing and social work for the care transition navigator. We have added that core components have been shown to be effective in increasing mental health treatment engagement in previous work (see pages 10-11). We now also provide a reference highlighting the evidence for the main outcome mental health treatment engagement for each of these three components (see page 11).

3. The CTI sessions will be provided via a web based teleconference system that is paid for. Won't this be a barrier for some patients who may not have the system or how to use it?

AUTHORS’ REPLY: The system will be paid for by the study team and is covered by the funding scheme. No fee will apply for patients. In routine care in Germany, the use of teleconferencing systems is covered by the statutory health insurance. We have added this detail to the manuscript on page 12.

4. Sufficiency and efficiency are some of the outcomes being measured. How will these two be analyzed?

AUTHORS’ REPLY: To estimate the sufficiency and efficiency of the recruitment strategies, we will calculate the recruitment yield, i.e., number randomized per number screened, and the consent rate, i.e., number randomized per number eligible. We have added this information to the manuscript on page 17.

Reviewer #3:

The authors address an important aspect of mental health services and provide a rationale for bridging inpatient and outpatient care, however , the rationale for the chosen illness conditions (Depression and Anxiety) is not provided. This makes it difficult to appreciate why CTI (the intervention ) should be the intervention of choice.

AUTHORS’ REPLY: CTIs have not been evaluated or even implemented in German routine mental healthcare. Indeed, in the most thorough reviews on transitional interventions (TI) provided by Vigod et al. (2013; https://doi.org/10.1192/bjp.bp.112.115030) and Tyler et al. (2019; https://doi.org/10.1186/s12913-019-4658-0) not a single trial comes from Germany. Hence, we selected depression and anxiety as highly prevalent mental health disorders with high burden of disease to pilot the CTI in German mental healthcare. We have added our considerations when planning the study on page 6.

Of note, CTI is a multicomponent intervention but the detail on how it has been developed , and what the effective components are (core and optional ) would help one to appreciate how CTI is likely to be effective. 

AUTHORS’ REPLY: We developed the CTI based on a thorough review of the literature (MEDLINE and Google Scholar with the search term “mental health treatment engagement" and no filters) finalized on February 8th, 2022. The literature was reviewed by a group of experts of different backgrounds (psychiatrist, epidemiologist, psychologist) Specifically, we selected the intervention components with respect to their potential effectiveness for increasing mental health treatment engagement. We have added the history of how the intervention was developed on page 9.

How will the patients be involved in the development of the intervention like is stated in the protocol (patient and public involvement) ?

AUTHORS’ REPLY: We have elaborated on how we will involve patients in the refinement of the intervention with respect to a large confirmatory trial (please see page 20).

A schematic or diagram illustrating CTI (and the treatment as usual for the control arm) would help explain the two study treatments under comparison.

AUTHORS’ REPLY: We have added a respective schematic, please see Figure 3.

A study conceptual framework should also be included in the protocol.

AUTHORS’ REPLY: We added subsection on the study conceptual framework including Table 1 which provides an overview.

It is not clear to me how patient recruitment will start in July 2022 and end in 2023 yet the protocol is under review in July 2023; 

AUTHORS’ REPLY: Thank you for pointing this out. Recruitment started in August 2022 and is on-going. We have amended the manuscript accordingly on page 6.

why is pregnancy in the 2nd and 3rd trimester part of the exclusion criteria? 

AUTHORS’ REPLY: For this pilot trial, we aimed for minimizing the rate of loss-to-follow-up. Hence, we excluded pregnant women who may not have been able to complete follow-up assessments due to giving childbirth. We will drop this exclusion criterion in the main trial. We have amended the manuscript accordingly on page 7.

And how will insufficient German language proficiency be determined?

AUTHORS’ REPLY: Before enrolling patients, we will regularly check with providers having delivered inpatient treatment if a therapeutic dialogue was possible and whether language proficiency had turned out as a problem during inpatient treatment. We have amended the manuscript accordingly on page 7.

With respect to the intervention; how will relapses be handled in the study?

AUTHORS’ REPLY: Patients experience relapses will have full and free access to all levels of mental healthcare which covered under the statutory health insurance in Germany. We will approach all patients systematically for postassessments regardless of whether they have received a relapse to minimize missing data for the follow-up. We have amended the manuscript accordingly on page 18.

How will the assessors who will collect data be blinded to the treatment allocation? this was not clear,

AUTHORS’ REPLY: Thank you for pointing this out. This is a mistake: In this feasibility trial, we will not blind outcome assessors. We have amended the manuscript accordingly.

The study seems like it will use a mixed methods approach but there is limited detail on qualitative methods, specifically the interviews; do they mean in depth interviews with patients and key informant interviews with the CTN? More elaboration on this aspect of the study is required including the qualitative data collection and analysis .

AUTHORS’ REPLY: Thank you for highlighting this aspect. This is correct. We will conduct semi-structured interviews both with patients and with the CTN. We now provide more details on the qualitative strand of our trial in the manuscript on page 19.

Reviewer #4:

The authors clearly and intelligibly wrote this manuscript in standard English. The rationale was stated well and it indeed is a worthwhile project. The protocol is technically sound and the methods used are replicable.

AUTHORS’ REPLY: Thank you for your positive evaluation of our work.

ISSUES to address

The links in the appendix on pages 30,31,32 and 33 all link to download of the SPIRIT guidelines. These should be redirected to the appropriate documents.

AUTHORS’ REPLY: We have redirected the links accordingly.

---

## [Editor Report · Decision Letter 1]

22 Aug 2023

From inpatient to outpatient mental health care: protocol for a randomised feasibility trial of a care transition intervention for patients with depression and anxiety (the AMBITION-trial)

PONE-D-23-11362R1

Dear Dr. Huan

We’re pleased to inform you that your manuscript has been judged scientifically suitable for publication and will be formally accepted for publication once it meets all outstanding technical requirements.

Kind regards,

Dickens Akena, Ph.D

Academic Editor

PLOS ONE
---

## [Editor Report · Acceptance letter]

25 Aug 2023

PONE-D-23-11362R1 

From inpatient to outpatient mental health care: Protocol for a randomised feasibility trial of a care transition intervention for patients with depression and anxiety (the AMBITION-trial) 

Dear Dr. Haun:

I'm pleased to inform you that your manuscript has been deemed suitable for publication in PLOS ONE. Congratulations! Your manuscript is now with our production department. 

Kind regards, 

on behalf of

Dr. Dickens Akena 

Academic Editor

PLOS ONE